# Modeling Cues May Reduce Sway Following Sit-To-Stand Transfer for People with Parkinson’s Disease

**DOI:** 10.3390/s23104701

**Published:** 2023-05-12

**Authors:** Rebecca A. Martin, George Fulk, Lee Dibble, Ali Boolani, Edgar R. Vieira, Jennifer Canbek

**Affiliations:** 1Department of Physical Therapy, Nova Southeastern University, Fort Lauderdale, FL 33314, USA; rmartin@wildfirept.com (R.A.M.);; 2Department of Rehabilitation Science, Emory University, Atlanta, GA 30322, USA; 3Department of Physical Therapy and Athletic Training, University of Utah, Salt Lake City, UT 84112, USA; 4Honors Program, Clarkson University, Potsdam, NY 13669, USA; 5Department of Physical Therapy, Florida International University, Miami, 33199 FL, USA

**Keywords:** Parkinson’s disease, cues, functional status, neurological rehabilitation

## Abstract

Cues are commonly used to overcome the effects of motor symptoms associated with Parkinson’s disease. Little is known about the impact of cues on postural sway during transfers. The objective of this study was to identify if three different types of explicit cues provided during transfers of people with Parkinson’s disease results in postural sway more similar to healthy controls. This crossover study had 13 subjects in both the Parkinson’s and healthy control groups. All subjects completed three trials of uncued sit to stand transfers. The Parkinson’s group additionally completed three trials of sit to stand transfers in three conditions: external attentional focus of reaching to targets, external attentional focus of concurrent modeling, and explicit cue for internal attentional focus. Body worn sensors collected sway data, which was compared between groups with Mann Whitney U tests and between conditions with Friedman’s Tests. Sway normalized with modeling but was unchanged in the other conditions. Losses of balance presented with reaching towards targets and cueing for an internal attentional focus. Modeling during sit to stand of people with Parkinson’s disease may safely reduce sway more than other common cues.

## 1. Introduction

Postural instability is one of the cardinal signs of Parkinson’s disease (PD) [1]. Long before people with PD develop a positive pull test, they exhibit increased postural sway during static standing in both the sagittal and coronal planes as compared to healthy age matched peers [2]. Changes in postural sway have been linked to fall risk within this population [2]. Additionally, when verbal-cognitive dual tasking is introduced, individuals with PD experience significantly more sway [2]. Because of the progressive neurodegeneration seen with PD [3], postural instability increases over time and impairs the ability of people with PD to complete motor tasks like ambulation and transfers.

In addition to increases in postural sway, individuals with PD experience akinesia, bradykinesia, tremor, rigidity, and lack of automaticity. Explicit cues, such as modeling, are one common intervention used to overcome the effects of many of these motor symptoms, though their effects on postural sway during discrete tasks are unclear. Functional magnetic resonance imaging has shown that explicit cues utilize neural circuits which bypass the basal ganglia [4]. Studies indicate that explicit cueing during gait can result in increased gait speed, [5,6] increased step length, [5,7] and decreased episodes of freezing during gait [8]. Some studies looked at the impact of continuous or discrete tasks in sitting, [9,10,11] but did not provide any insight into the impact of cues on postural sway. Other studies have demonstrated that type of cue is important with some external cues improving movement, whereas others degrade the movement [12].

Some studies have examined the impact of providing explicit cues during the task of sit-to-stand (STS) transfers on the motor control of people with PD [13,14]. One study reported that visual and auditory cues may result in the decreased duration of the transfer, improved peak horizontal and vertical velocities, and improved joint torque time to peak [13]. Another study reported a shorter duration of transfer and improved hip flexion torque after four weeks of training with explicit cues [14]. Both of these studies provide support for the use of explicit cues to improve the motor control of people with PD during the transfer. However, it is also important to understand the impact of these cues on postural control in immediate standing balance.

To the best of the authors’ knowledge, no studies have examined the effect of cues provided during the STS transfer on postural control during immediate standing in people with PD. Identifying the potential positive or negative impact of commonly applied cues on postural control will allow clinicians to make informed decisions regarding selection of cues and intervention designs. The purpose of this study was to identify whether three different types of explicit cues provided during STS transfers of people with PD led to postural control during immediate standing that was more similar to healthy controls. It was hypothesized that explicit cues which elicited an external attentional focus, modeling or reaching to targets, would result in improved postural control, whereas the cue that elicited an internal attentional focus would result in reduced postural control.

## 2. Materials and Methods

### 2.1. Study Sample

A PD group and a healthy control (HC) group, each with 13 subjects, were recruited from exercise and support groups throughout central and northern New York. Inclusion criteria were: stable on medications for the past two months, able to follow directions, able to rise from a chair without assistance or use of their arms at least one of every four attempts during uncued testing, be between the ages of 45 and 90, and score a minimum of 22 on the Montreal Cognitive Assessment (MoCA) [15]. In addition, those in the PD group were required to have been diagnosed with idiopathic PD by a neurologist. Potential subjects were excluded from the study if they had brain surgery for the treatment of PD, were participating in a medication study, had a body mass index greater than 35 as this could significantly alter the mechanics of an STS transfer, [16,17] or had any other health conditions that would impair their ability to complete an STS transfer. Institutional Review Board approval was attained from Clarkson University and Nova Southeastern University. All subjects signed an informed consent form prior to participation in the study.

### 2.2. Study Design

A cross-over design was used in which all subjects in the experimental group completed each condition of the STS transfer. The HC group completed only the uncued STS condition to allow for comparison of postural sway of the PD group across conditions against neurologically healthy controls.

### 2.3. Demographic Measures

To determine the profile of the sample, age, gender, height, weight, and gait speed were collected from all subjects. Gait speed was measured via the 10 Meter Walk Test, which has been found to be valid and reliable within healthy adults and adults with PD [18]. Additionally, those in the PD group completed the Parkinson’s Disease Questionnaire 39 [19,20] and were tested by a trained rater on the Movement Disorder Society’s Unified Parkinson’s Disease Rating Scale [19,21]. These tests and measures have been found to be valid and reliable in adults with PD to assess the impact of PD on quality of life and disease severity.

### 2.4. Equipment and Dependent Variables

Sway data were collected during all STS trials utilizing an inertial measurement unit sensor (Opal Sensors and Moveo Explorer Data Collection Program, APDM Wearable Technologies, Portland, OR, USA) placed on the lumbar spine using an elastic strap around the trunk. This sensor utilized gyroscopes, magnetometers, and accelerometers to collect sway data during the first 30 s of standing following an STS transfer. These sensors have been found to be valid and reliable in both healthy adults [22,23] and individuals with PD [23,24,25]. The unit communicated with a nearby computer to collect data in real time. Sway data collected included sway area, amount of sway in the sagittal plane (see Figure 1), amount of sway in the coronal plane (see Figure 1), sway jerk, and sway velocity. Sway area was defined as 95% of the total area of the individual sways and combined movement in both the sagittal and coronal planes [26]. Sway in the sagittal and coronal planes was defined as the total degrees of sway experienced in the related plane [26]. Sway jerk was a derivative of acceleration and provides information regarding the smoothness of movements [22]. Sway velocity was the average speed of sway movements [26]. Additionally, any incidents of the center of mass (COM) moving outside of the base of support (BOS), as demonstrated by the subject stepping or requiring assistance to prevent a fall, were operationally defined as a loss of balance (LOB), and the number was recorded.

All data were collected in a private location in a university setting or a similar room in a community exercise facility during a single session. In order to ensure that all data for those in the PD group was collected during the same phase of medication, all motor data was collected in a 60-min window that began exactly 60 min following the consumption of their regularly scheduled dopamine replacement therapy [27]. This timing would result in data collection during peak “on” time [27].

Both the HC and PD groups completed three trials of uncued STS trials. All STS trials began from a standardized position. Subjects were seated on a tub bench with the seat height adjusted such that their knees were in 100° of flexion when their tibia were vertical. Then, the subjects’ feet were placed shoulder width apart and moved posteriorly 10 cm to place the subjects’ feet in the position most optimal for completing the STS transfer. Subjects were provided with a verbal prompt requesting that they respond in a single sentence and then stand immediately. The subjects were required to provide a verbal response prior to standing so that they could not use a cue from the tester to initiate their sit-to-stand motion. Sway data was collected during the first 30 s of standing using the inertial measurement unit sensor. In addition, subjects were monitored for incidents of their COM moving outside the BOS that resulted in a step or required physical assistance to recover.

Those in the PD group additionally completed three STS trials in each of the three conditions. The nine experimental trials were completed in a random order (Randomizer.org, Social Psychology Network, Middletown, Connecticut). Prior to each trial, the tester read a condition-specific set of directions that asked the subject to focus on the current set of directions. Condition 1 (Modeling for External Attentional Focus): Each subject began in the standardized position with a second tub bench placed opposite the subject at a distance from the subject’s toes equal to two and a half times the length of the subject’s arm. The tester provided a verbal prompt of “when I stand up, stand with me” and then the tester stood up. Condition 2 (Reaching to Targets for External Attentional Focus): Each subject began in the standardized position with a second tub bench placed opposite the subject at a distance from the subject’s toes equal to the length of the subject’s arm. The tester provided the prompt to “reach to my hand” as the tester placed the dorsum of their hand on the front of the opposing chair. When the subject contacted the tester’s hand, the test immediately cued the subject to “stand to the ceiling.” Condition 3 (Cued for Internal Attentional Focus): Each subject began in the standardized position. All objects were removed from in front of the subject. The tester provided the cue to “bend forward at your hips and stand until your back is straight”.

### 2.5. Data Analysis

Data were analyzed using SPSS Version 26.0. Demographics were calculated for both the PD and HC groups. Each dependent variable was assessed for normality utilizing the Shapiro-Wilk Test. Results were assessed for skewness and kurtosis. Sway data was graphed and reviewed looking for indications of learning or fatigue, such as trend lines suggesting improvement or decline throughout the course of trials. No trends were identified.

### 2.6. Comparison of Postural Sway between the PD and HC Groups

Sway metrics observed during the uncued condition were compared between the PD and HC groups using Mann-Whitney U tests with a significance level of *p* < 0.013 (0.5/4). For those sway metrics with significant differences found between the PD and HC groups during the uncued conditions, separate Mann-Whitney U tests were completed between the uncued condition of the HC group and each experimental condition of the PD group. This was completed to identify if any experimental condition resulted in postural sway characteristics that were not significantly different from the HC group.

### 2.7. Comparison of Postural Sway across Conditions for the PD Group

Separate Friedman’s tests were utilized to compare the postural sway area, sagittal plane sway, coronal plane sway, sway jerk, and sway velocity of the PD group across all conditions of the STS transfers (uncued, modeling, reaching, and internal focus). Because we performed five separate Friedman tests (one for each measure of postural control), Bonferroni correction factors were applied, resulting in a *p* < 0.01 (0.05/5) for statistical difference. If a significant difference was identified in Friedman’s test, a post-hoc Mann-Whitney U test was performed to identify which cueing conditions caused the difference. A Bonferroni correction factor was applied resulting in *p* < 0.008 (0.5/6) because there were six potential comparisons. A McNemar Test (Bonferroni corrected *p* < 0.008, (0.05/6) was utilized to compare the occurrences of COM moving outside of the BOS across conditions.

## 3. Results

Subjects in the PD group included eight males and five females with a mean age of 68.46 (±9.11) years and a gait speed of 0.87 (±0.21) m/s. Subjects in the HC group included seven males and six females with a mean age of 67.31 (±10.41) and a gait speed of 1.23 (±0.12) m/s. One candidate for the PD group was excluded from the study for not meeting the minimum criteria during cognitive testing. Additional demographic information can be found in Table 1.

### 3.1. Comparison of Postural Sway between the PD and HC Groups

There was a statistically significant difference (*p* < 0.013) between the PD and HC groups during the uncued condition with those in the PD group demonstrating increased sway area (PD = 5.192°^2^, HC = 0.767°^2^) and sway jerk (PD = 5.03 m/s^3^, HC = 0.644 m/s^3^). No significant difference was noted between the PD and HC groups during the uncued condition for coronal sway, sagittal sway, or sway velocities. When the sway area was compared between the uncued condition of the HC group and the experimental conditions of the PD group, a significant difference remained across all conditions. However, when sway jerk was compared between the uncued condition of the HC group and the experimental conditions of the PD group, the significant difference was no longer present in the modeling condition (PD = 2.361 m/s^3^, HC = 0.644 m/s^3^) (see Table 2). Additionally, no LOB occurred for either group in the uncued condition.

### 3.2. Comparison of Postural Sway across Conditions for the PD Group

There was a statistically significant difference between conditions (*p* < 0.01) for the PD group in coronal sway. Post-hoc testing identified that the modeling cue resulted in significantly less coronal sway than the uncued, reaching to target, or internal attentional focus conditions (uncued = 0.272°, reach to targets = 0.265°, modeling = 0.197°, internal attentional focus = 0.288°) (see Table 3). Additionally, LOB occurred during both the reaching to target and internal attentional focus conditions. Two LOB incidents occurred during the reaching to target condition, one which was self-corrected with a step and one which required tester assistance to recover. One LOB incident occurred during the internal attentional focus condition and the subject required tester assistance to recover. A McNemar Test found no statistical significance (*p* < 0.008) regarding episodesof LOB between conditions for the PD group (see Table 4).

## 4. Discussion

This study examined the effects of three different types of cueing on postural sway immediately following an STS transfer for people with PD who experience occasional difficulty completing an STS from a standard-height chair. When not cued, individuals with PD were found to have significantly greater sway areas and sway jerks in immediate standing than their healthy counterparts. A verbal cue paired with a modeling cue resulted in decreased postural sway during the first 30 s of standing without LOB. Neither a verbal cue paired with reaching targets nor a verbal cue for internal attentional focus decreased postural sway during the first 30 s of standing. However, both cues introduced LOB incidents that were not present during the uncued condition for either the HC or PD groups. Our study provides insight into the clinical application of cues utilizing modeling, target, and internal attentional cues during STS transfers and the impact of these cues on postural stability.

### 4.1. Modeling

An explicit verbal cue paired with modeling of the STS transfer was the only cue that resulted in improved balance during early standing for the PD group. This suggests that, especially for individuals with clinically important postural instability, modeling may be a safe way to cue individuals while completing standing tasks. Prior research suggested that increases in sway jerk may be the best sway characteristic to identify untreated PD [22]. In our study, modeling was able to reduce the level of sway jerk to not significantly different from the HC group.

In this study, modeling was completed in “mirror image” positioning with the tester directly across from the subject. However, this may not be possible in all environments. Consideration of the theoretical basis for why modeling was effective suggests that replicating the paired verbal command and modeling from beside the patient, or even utilizing a squatting position without a seat, should provide similar results. Modeling with a paired verbal command provides an explicit cue, which functional magnetic resonance imaging [5] has shown to utilize neural pathways that are not reliant on the basal ganglia. Additionally, electromyography (EMG) studies have shown that, even in the absence of movement, similar activation occurs in the motor cortex when observation of familiar movements occurs [28]. This activation is known as the mirror neuron system and explains how humans can predict what happens next during a familiar sequence of events. [28] Research indicates that action observation, like watching a sit-to-stand transfer, could activate the mirror neuron system and prime the motor cortex for improved movements [29].

### 4.2. Incidents of Loss of Balance

While no statistically significant changes in sway area, coronal sway, sagittal sway, sway jerk, or sway velocity were found when subjects completed the reaching to target or internal attentional focus conditions, incidents of LOB were present in both conditions. A McNemar Test indicated that the incidents of LOB were not statistically significant. However, a Delphi study reported that a 25% decrease in falls should be considered a significant improvement following interventions for people with PD [30]. While the authors of this study did not provide an operational definition for “fall”, it is likely the loss of balance which required tester assistance to recover would have fit within their operational definition. It is less likely that the self-initiated step would fit into this definition. If we considered the sample as a whole, the percentage of LOB incidents by condition would still be under this criterion. However, with all three LOB incidents occurring with different subjects, it is important to note that based on the suggested minimal clinically important difference (MCID), two to three different subjects may have experienced clinically important increases in LOB incidents with the addition of an explicit cue. Two out of 13 subjects may have experienced a clinically meaningful increase in LOB incidents within the reaching-to-target cue. One out of the 13 subjects likely experienced a clinically meaningful increase in LOB incidents within the reaching-to-target cue.

### 4.3. Selection and Placement of Targets

Our findings of the introduction of LOB incidents suggest that targets should be carefully selected to reduce the risk of falls. This is consistent with other research suggesting that tape lines on the floor were more effective targets than wearable laser lights for providing cues to improve gait [31]. During the reaching to target condition, individuals were cued to reach to the tester’s hand, which occurred at the end of the pre-extension phase, then to “stand to the ceiling” to complete the extension phase. All subjects were able to reach the tester’s hand without signs of imbalance. However, the two LOB incidents that occurred during the reaching-to-target condition occurred at the end of the extension phase, which could indicate that the ceiling was not an appropriate target.

Prior research has reported improved motor control during discrete tasks completed by people with PD with the introduction of targets, but did not add to our understanding of the impact of utilizing targets on postural sway and balance since studied reaching tasks were completed in one sitting [9,10]. To the best of these authors’ knowledge, this is the first study to report on the impact of reaching-to-targets on postural sway. Based on prior research [10,31] and the additional findings within the current study, targets are likely an effective strategy to improve motor control for individuals with PD, but clinicians should strive to place targets in attainable locations that result in optimal movement. In the case of the STS transfer, it may have been better to have the tester place a hand at shoulder height for the subject to reach when standing. In the clinic, if a therapist is seeking to improve upper extremity swing during a step and reach activity, it may be better to place a target at the maximally attainable distance than to encourage the patient to “reach toward that wall”.

### 4.4. Cueing for an Internal Attentional Focus

Despite frequent use in clinical and home settings of the cue used in this study for an internal attentional focus, or similar phrases, this study does not support the clinical utility of such cues for this population. No benefits were seen regarding improvements in postural sway. Rather, one LOB incident was introduced with the addition of this cue, indicating that it may reduce the safety and independence of a sit-to-stand transfer in this population.

### 4.5. Limitations and Future Research

The sample size of this study, while adequate to find significance, is still relatively small and represents a limited range of disease severity. At this time, it is unclear if these findings would apply to individuals with more severe cognitive or motor impairments due to PD. Many different cues are provided in the clinic; however, in this study, we only looked at three cues. Therefore, there may be other cues that are more effective at improving the postural stability of people with PD in early standing that were not examined here. Lastly, this study looked at the effects of a one-time cue. Further research should examine the effects of practice on skill acquisition and retention of cued STS transfers.

### 4.6. Clinical Implications

Modeling providing a succinct verbal cue may improve postural control while completing discrete standing tasks provides caregivers and clinicians with a useful cue that can be provided in most settings. With an understanding of the theoretical basis of modeling, caregivers and clinicians should complete modeling cues from any location that allows for people with PD to clearly see what is being modeled and maximizes safety for both individuals.

## Figures and Tables

**Figure 1 sensors-23-04701-f001:**
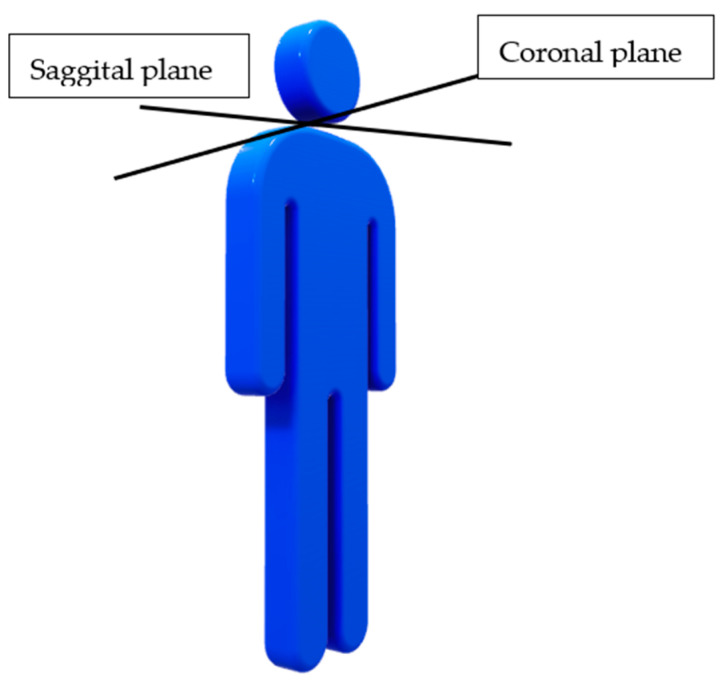
Sagittal and Coronal Planes.

**Table 1 sensors-23-04701-t001:** Subject Demographics.

Characteristic	Healthy Control (*n* = 13)	Subjects with PD (*n* = 13)	95% CI
Gender (male/female)	7/6	8/5	
Age in years Mean (SD)	67.31 (10.41)	68.46 (9.11)	−9.07, 6.77
Height in cm Mean (SD)	165.72 (10.49)	172.00 (8.72)	−14.09, 1.52
Weight in kg Mean (SD)	82.36 (14.41)	84.73 (12.54)	−13.69, 8.95
10MWT (m/s) Mean (SD)	1.23 (0.12)	0.87(0.21)	−3.09, −1.03
Years with symptoms Mean (SD)		10.38 (9.18)	-
Years with diagnosis Mean (SD)	-	5.38 (3.3)	-
MDS-UPDRS–total score–Median (range) Possible range: 0–199	-	70 (48–112)	-
PDQ-39–Median (range) Possible range: 0–100	-	34 (4–74)	-

10 MWT = 10 Meter Walk Test; MDS-UPDRS = Movement Disorder Society’s Unified Parkinson’s Disease Rating Scale; PDQ-39 = Parkinson’s Disease Questionnaire 39.

**Table 2 sensors-23-04701-t002:** Comparing Sway Characteristics Across Parkinson’s Disease Conditions to the Healthy Control Uncued Condition.

Postural Sway Characteristic	Healthy Control Mean (SD)	Parkinson’s Disease Mean (SD)			
	Uncued	Uncued	Modeling	Reach to Targets	Internal Focus
Sway Area (°^2^)	0.767 (0.303)	5.192 (7.074) *	3.147 (3.893) *	4.824 (7.100) *	4.316 (4.798) *
Coronal Sway (°)	0.158 (0.103)	0.272 (0.177)	0.197 (0.141)	0.265 (0.200)	0.288 (0.218)
Sagittal Sway (°)	0.445 (0.248)	0.816 (0.724)	0.542 (0.340)	0.702 (0.425)	0.601 (0.396)
Sway Jerk (m/s^3^)	0.644 (0.417)	5.030 (7.28) *	2.361 (3.061)	2.564 (2.69) *	2.320 (2.501) *
Sway Velocity (m/s)	0.149 (0.153)	0.215 (0.178)	0.157 (0.141)	0.204 (0.181)	0.207 (0.143)

* Sway characteristic of the PD group is significantly different than the HC group after the Bonferroni correction (*p* < 0.013).

**Table 3 sensors-23-04701-t003:** Sway Characteristics Compared Across Conditions in Parkinson’s Disease Group.

Sway Characteristic	Condition Mean (SD)			
	Uncued	Modeling	Reach to Targets	Internal Attentional Focus
Sway Area (°^2^)	5.192 (7.074)	3.147 (3.893)	4.824 (7.100)	4.316 (4.798)
Coronal Sway * (°)	0.272 (0.177) *†*	0.197 (0.141) *†*	0.265(0.200) *†*	0.288 (0.218)
Sagittal Sway (°)	0.816 (0.724)	0.542 (0.340)	0.702 (0.425)	0.601 (0.396)
Sway Jerk (m/s^3^)	5.030 (7.28)	2.361 (3.061)	2.564 (2.69)	2.320 (2.501)
Sway Velocity (m/s)	0.215 (0.178)	0.157 (0.141)	0.204 (0.181)	0.207 (0.143)

* Statistical significance was found for this postural sway characteristic through Friedman’s test after Bonferroni correction (*p* < 0.01) *†* Statistical significance was found with post-hoc Mann Whitney U after the Bonferroni correction (*p* < 0.008). Modeling resulted in significantly less coronal plane sway than the uncued or reaching target conditions.

**Table 4 sensors-23-04701-t004:** Table of Losses of Balance **.

Losses of Balance	Uncued	Reaching to Target	Modeling	Internal Attentional Focus
Step	-	1	-	-
Assistance	-	1	-	1

COM = center of mass, BOS = base of support; No statistical significance was found after Bonferroni correction (*p* < 0.008) ** Loss of balance operationally defined as the COM moving outside the BOS.

## Data Availability

Not applicable.

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
