# Peer review of "Modeling Cues May Reduce Sway Following Sit-To-Stand Transfer for People with Parkinson’s Disease"

_sensors, 2023, doi:10.3390/s23104701_

Round 1

Reviewer 1 Report

This study was to examine whether the modeling cues may reduce sway following sit to stand transfer for people with Parkinson disease or not. Feedback as a positive effector for therapeutic approaches is very important thing, so this study is important research for clinical setting.

I have a question why the researcher designed the experiment process differently for the two groups, Parkinson’s disease and Healthy control. This study mentions the reasons from line 87 to 89 in the manuscript (The HC group completed only the uncued STS condition to allow from comparison of postural sway of the PD group across conditions against neurologically healthy controls.). If it’s judged that the cue is important for postural stability and it’s intended to be applied as a method of therapeutic intervention, it’s necessary to compare healthy control and PD when the uncued sit-to-stand condition is given, as well as a comparison between healthy control and PD when the cue sit-to-stand is given.

 Second question is what is the modeling cues? Title of this study is “Modeling cues may reduce sway following sit-to-stand transfer for people with PD.” Modeling cues are important parameter, independent parameter in this study. However, the introduction section did not involve any description of the modeling cues or mirror image.

 Third question is what do you think there was a significant different in sway area when comparing 5 different measure conditions (uncued condition in healthy control, and uncued, reach-to-targets, modeling, and internal attentional focus conditions in PD), but not when comparing only 4 conditions (uncued, reach-to-targets, modeling, and internal attentional focus conditions in PD). On the other hand, why was there a significantly difference in coronal sway when comparing 4 different measure conditions, but not when comparing 5 different measure conditions. Statistics values other than the mean and standard deviation should be provided to help the reader understand properly.

Miner comments.

Abstract

1. Methods: Please add the common characteristics such as age, sex of stroke and control group.

2. Results: It is desirable to insert numbers such as mean, standard deviation, median, IQT, p-value, in the results’ description to facilitate the reader’s understanding.

Minor editing of English language required

Author Response

Thank you to all who have participated in our review process so far. We found the feedback from the reviewer helpful and reflected carefully on each item. Our responses, explanations, and corrections are outlined below and shown in tracked changes throughout the document. Our responses to the reviewer are in red.

Thank you for the opportunity to revise this manuscript.

Reviewer 1-

I have a question why the researcher designed the experiment process differently for the two groups, Parkinson’s disease and Healthy control. This study mentions the reasons from line 87 to 89 in the manuscript (The HC group completed only the uncued STS condition to allow from comparison of postural sway of the PD group across conditions against neurologically healthy controls.). If it’s judged that the cue is important for postural stability and it’s intended to be applied as a method of therapeutic intervention, it’s necessary to compare healthy control and PD when the uncued sit-to-stand condition is given, as well as a comparison between healthy control and PD when the cue sit-to-stand is given.

Thank you for your question. It was reviewed and appreciated, but no changes were made to the manuscript.

The goal of the study was to determine which type of cue result in the most “normal” sit to stand transfer. Therefore, we only included the healthy control group to determine what “normal” sit to stand findings would be in our controlled environment. Knowing how the healthy controls respond to the cues would not help us to answer our research question.

If we did not have the “normal” measured with healthy controls, then if we found a significant change in response to a cue we would not know if the cue resulted in a more “normal” or less “normal” response.

If we compared the performance of the individuals with Parkinson disease to the performance of a healthy control group responding to a cue that would not tell us if the individual with Parkinson disease had a more “normal” sit to stand transfer.

 Second question is what is the modeling cues? Title of this study is “Modeling cues may reduce sway following sit-to-stand transfer for people with PD.” Modeling cues are important parameter, independent parameter in this study. However, the introduction section did not involve any description of the modeling cues or mirror image.

We are not aware of any current research on the impact of modeling cues specific to Parkinson disease so this could not be included in the introduction. However, we spoke more broadly of the known impact of “explicit cues” since modeling is one type of explicit cue. Clinically, modeling is often used so we chose this an explicit cue to explore.

We made the following change to the manuscript: we changed line 41 to clarify that modeling is a type of explicit cue.

 Third question is what do you think there was a significant different in sway area when comparing 5 different measure conditions (uncued condition in healthy control, and uncued, reach-to-targets, modeling, and internal attentional focus conditions in PD), but not when comparing only 4 conditions (uncued, reach-to-targets, modeling, and internal attentional focus conditions in PD).

This was expected and is due to the postural control problems experienced by people with Parkinson disease.

Table 2 shows the results of comparison of postural sway between those with Parkinson disease and those in the healthy control group.Rather than comparing all 5 columns, you will note that we compared each condition for the people with Parkinson disease to the performance of a “typical” stand as measured in the healthy control group. Table 2 shows that in all conditions the sway area of the Parkinson disease group was significantly different than the healthy controls. This indicates that in all conditions, the individual with Parkinson disease had significantly greater sway than the healthy controls. This was expected and is theorized to occur due to changes in automaticity resulting from basal ganglia dysfunction that occurs in Parkinson disease.

In table 3, the healthy control were not included, since we were comparing the performance of the individuals with Parkinson disease across the conditions.  The sway area was not significantly different between conditions indicating that the sway area was not significantly altered by the type of explicit cue provided.

On the other hand, why was there a significantly difference in coronal sway when comparing 4 different measure conditions, but not when comparing 5 different measure conditions.

This occurred because there was an improvement in sway in the coronal plane for individuals with Parkinson disease between the uncued condition and their performance with modeling or reaching to target cues.  

When comparing a single condition to the healthy controls, as seen in Table 2, sagittal sway was not found to be significantly different than the healthy controls. However, in table 3, the healthy control were not included as we were comparing the performance of the individuals with Parkinson disease in the different conditions. Then, we did see a significant decrease, or improvement, in sway in the coronal plane. This means that the individuals with Parkinson disease swayed less side to side when they had a modeling or reaching to target cue.

Statistics values other than the mean and standard deviation should be provided to help the reader understand properly.

Because we are comparing the average performance between conditions or between the performance of a condition and the “normal” as represented by the healthy controls, we provided the mean and standard deviation. We believe that this is appropriate because it is ratio level data.

Miner comments.

Abstract

  1. Methods: Please add the common characteristics such as age, sex of stroke and control group.

This paper was about Parkinson disease, so we included the gender and age characteristics for the Parkinson group and the healthy control group in the table titled “Table 1. Subject Demographics.” The table can be found on line 187. However, character limitations preclude us from including this in the abstract itself. If the reviewer would like we can work with the editorial staff to correct that.

  1. Results: It is desirable to insert numbers such as mean, standard deviation, median, IQT, p-value, in the results’ description to facilitate the reader’s understanding.

As appropriate, we included this information within the results section of the paper. However, character limitations preclude us from including this in the abstract itself.

Reviewer 2 Report

This interesting paper addresses a research gap that is relevant to researchers, caregivers, and clinicians regarding the clinical application of cues during sit-to-stand transfer and their impact on postural stability for people with Parkinson's disease. The main strengths of this research work are its methodology, results, and discussion sections. Furthermore, the article's contents are well-organized and written clearly.

However, authors can make some minor revisions to improve the final version of this paper for publication:

1 – Lines 25-26: In different sections of the paper, such as section 3.2 and section 4.2, the authors use the term "Lost of Balance" (LOB) to describe the negative effects detected when using reaching to target and internal attentional focus conditions. In the Abstract, the term used to describe these negative effects is the following: "incidences of the center of mass moving outside the base of support". I recommend that the authors use the same term throughout the paper, and in my opinion, the term "Lost of Balance" is easier for readers to understand.

2 – Line 69: At the end of this line a number 2 appears but seems to be a typographic error. Please delete it.

3 – Lines 100 to 116: "In section 2.4, a figure showing the different planes and measures described in this section can help the reader to understand its contents more easily and quickly.

4 – Line 102: In this sentence, the authors explained that they placed the IMU sensor on the lumbar spine of the subjects who participated in the experiment. Could you also please explain how the sensors were attached to the participants' bodies?

5 – Lines 136 to 138: The authors explained that they completed the nine experimental trials developed with the PD group in a random order. Could you explain why they chose to do so? Is there any literature that suggests random order has advantages over following the same order in all experiments?

6 – Tables 2 and 3: The order used by the authors in Tables 2 and 3 for the different types of cues tested in this research is different from the order used in section 2.5 to explain each of the three different conditions. Please follow the same order, as it will help readers understand the results of your research more easily.

7- Line 242: The first line of section 4.1 is repeated, please delete one.

I hope that these recommendations are useful to you and that your research, which will contribute to the advancement of clinical practice and improve the quality of life of Parkinson's patients, will soon be published in this journal.

Author Response

Thank you to all who have participated in our review process so far. We found the feedback from the reviewer helpful and reflected carefully on each item. Our responses, explanations, and corrections are outlined below and shown in tracked changes throughout the document. Our responses to the reviewer are in red.

Thank you for the opportunity to revise this manuscript.

Reviewer 2-

This interesting paper addresses a research gap that is relevant to researchers, caregivers, and clinicians regarding the clinical application of cues during sit-to-stand transfer and their impact on postural stability for people with Parkinson's disease. The main strengths of this research work are its methodology, results, and discussion sections. Furthermore, the article's contents are well-organized and written clearly.

We are glad you value our work. Thank you.

However, authors can make some minor revisions to improve the final version of this paper for publication:

1 – Lines 25-26: In different sections of the paper, such as section 3.2 and section 4.2, the authors use the term "Lost of Balance" (LOB) to describe the negative effects detected when using reaching to target and internal attentional focus conditions. In the Abstract, the term used to describe these negative effects is the following: "incidences of the center of mass moving outside the base of support". I recommend that the authors use the same term throughout the paper, and in my opinion, the term "Lost of Balance" is easier for readers to understand.

This was corrected throughout the paper at lines 25/26 and 267/268.

2 – Line 69: At the end of this line a number 2 appears but seems to be a typographic error. Please delete it.

Thank you for pointing this out. We have corrected it.

3 – Lines 100 to 116: "In section 2.4, a figure showing the different planes and measures described in this section can help the reader to understand its contents more easily and quickly.

We have added a 3D clipart to display the coronal and sagittal planes.

4 – Line 102: In this sentence, the authors explained that they placed the IMU sensor on the lumbar spine of the subjects who participated in the experiment. Could you also please explain how the sensors were attached to the participants' bodies?

We added “using an elastic strap around the trunk.”

5 – Lines 136 to 138: The authors explained that they completed the nine experimental trials developed with the PD group in a random order. Could you explain why they chose to do so? Is there any literature that suggests random order has advantages over following the same order in all experiments?

Thank you for giving us the opportunity to explain this protocol. Literature supports that individuals with Parkinson disease have the best motor learning during blocked practice due to damage of the basal ganglia. The basal ganglia is key to motor learning. Randomizing the order of the trials will greatly reduce their ability to improve across the three trials in each condition. Data from Lin and colleagues (2007) suggests that a block design is significantly better than a randomized design.

Lin, C. H., Sullivan, K. J., Wu, A. D., Kantak, S., & Winstein, C. J. (2007). Effect of task practice order on motor skill learning in adults with Parkinson disease: a pilot study. Physical therapy, 87(9), 1120-1131.

6 – Tables 2 and 3: The order used by the authors in Tables 2 and 3 for the different types of cues tested in this research is different from the order used in section 2.5 to explain each of the three different conditions. Please follow the same order, as it will help readers understand the results of your research more easily.

This is a great recommendation. Thank you. We have made the necessary changes

7- Line 242: The first line of section 4.1 is repeated, please delete one.

We have corrected this.

I hope that these recommendations are useful to you and that your research, which will contribute to the advancement of clinical practice and improve the quality of life of Parkinson's patients, will soon be published in this journal.

We’d like to thank the reviewer for their time. Their comments were extremely helpful. We that the manuscript is significantly stronger due to the reviewer’s feedback.

Round 2

Reviewer 1 Report

Thank you for your corrected manuscript.